# Relationship between Epithelial-to-Mesenchymal Transition and Tumor-Associated Macrophages in Colorectal Liver Metastases

**DOI:** 10.3390/ijms232416197

**Published:** 2022-12-19

**Authors:** Aurora Gazzillo, Michela Anna Polidoro, Cristiana Soldani, Barbara Franceschini, Ana Lleo, Matteo Donadon

**Affiliations:** 1Hepatobiliary Immunopathology Laboratory, IRCCS Humanitas Research Hospital, 20089 Rozzano, MI, Italy; 2Department of Biomedical Sciences, Humanitas University, 20072 Pieve Emanuele, MI, Italy; 3Division of Internal Medicine and Hepatology, Department of Gastroenterology, IRCCS Humanitas Research Hospital, 20089 Rozzano, MI, Italy; 4Department of Health Sciences, Università del Piemonte Orientale, 28100 Novara, NO, Italy; 5Department of General Surgery, University Maggiore Hospital Della Carità, 28100 Novara, NO, Italy

**Keywords:** colorectal liver metastases, epithelial-to-mesenchymal transition, tumor-associated macrophages, TGF-β signaling

## Abstract

The liver is the most common metastatic site in colorectal cancer (CRC) patients. Indeed, 25–30% of the cases develop colorectal liver metastasis (CLM), showing an extremely poor 5-year survival rate and resistance to conventional anticancer therapies. Tumor-associated macrophages (TAMs) provide a nurturing microenvironment for CRC metastasis, promoting epithelial-to-mesenchymal transition (EMT) through the TGF-β signaling pathway, thus driving tumor cells to acquire mesenchymal properties that allow them to migrate from the primary tumor and invade the new metastatic site. EMT is known to contribute to the disruption of blood vessel integrity and the generation of circulating tumor cells (CTCs), thus being closely related to high metastatic potential in numerous solid cancers. Despite the fact that it is well-recognized that the crosstalk between tumor cells and the inflammatory microenvironment is crucial in the EMT process, the association between the EMT and the role of TAMs is still poorly understood. In this review, we elaborated on the role that TAMs exert in the induction of EMT during CLM development. Since TAMs are the major source of TGF-β in the liver, we also focused on novel insights into their role in TGF-β-induced EMT.

## 1. Background

Colorectal cancer (CRC) is the third most commonly diagnosed malignancy, as well as the second leading cause of cancer-related mortality worldwide. It is the second most frequent cancer in women, after breast cancer, and the third most frequent in men, following lung and prostate cancer [1]. Although the incidence and mortality of CRC is showing a steady decrease in population over age 50 due to effective cancer screening measures, the occurrence of this malignancy in people younger than 50 years has been rapidly rising over the past 10 years [2]. Despite significant improvements in its diagnosis and treatments, most deaths are due to the development of distant metastasis characterized by a highly resistance to conventional therapies [3,4]. Indeed, 25–30% of CRC patients already have colorectal liver metastasis (CLM) at the time of diagnosis, or they will develop them after the resection of the primary tumor with an extremely poor 5-year survival rate [5].

In the last years, the crosstalk between cancer cells and the tumor microenvironment (TME) has been shown to play a key role in the clinical outcomes of CRC patients [6,7]. Indeed, tumor-associated macrophages (TAMs), the most abundant cells in TME, have been shown to promote tumor cells invasion and extravasation, to provide a supportive microenvironment for metastases and to be key determinants for the efficacy of anticancer strategies [8,9]. Among the pro-tumor mechanisms promoted by TAMs, TGF-β signaling represents a powerful activator of epithelial-to-mesenchymal transition (EMT), a process that plays a crucial role in CRC metastasis and in the resistance to chemotherapy and immunotherapy drugs [10,11,12,13]. Notably, the EMT represents an interesting therapeutic target in the treatment of cancer and could be exploited either to prevent tumor dissemination in patients with high risk to develop metastatic lesions or to eradicate existing metastatic cancer cells in patients with advanced disease [14]. Furthermore, despite it is well-recognized that the crosstalk between tumor cells and the inflammatory microenvironment is crucial in the EMT process, the association between the EMT and the role of TAMs is still poorly understood [15].

In this light, we reviewed the current knowledge about the EMT in CRC invasion, focusing on the role that TAMs play in the development of CLMs by inducing EMT through the TGF-β signaling pathway.

## 2. The Role of Epithelial-to-Mesenchymal Transition (EMT) in CRC

The liver is the most common CRC metastatic site as it receives and filter the blood from the intestine through the portal vein [16]. CLMs represent the final stage of a multi-step biological process. Firstly, cancer cells begin to migrate to the surrounding tissues near the primary CRC site and then they spread in venules, capillaries and lymphatic vessels until they enter the systemic circulation [17]. Once in the vasculature, circulating tumor cells (CTCs) reach the sinusoidal vessels of the liver [18]. The EMT mechanism provides tumor cells with several dynamic properties that help them to overcome environmental selective limitations of the metastatic translocation [19]. Upon EMT activation, tumor cells undergo to a wide range of biological and molecular changes, which facilitate their dissemination from the primary site, to the formation of metastases in distant organs [20]. Morphologically, EMT leads to the loss of typical polygonal shape of the cancer epithelial cells and the emergence of a more spindle-shaped fibrous mesenchymal-like cells. Epithelial cells are characterized by tight junctions, the expression of intercellular adhesion molecules and apical–basal polarity [21,22]. During EMT, they downregulate the expression of several epithelial proteins, such as Epithelial-cadherin (E-cadherin), claudins, and occludins, while upregulating the mesenchymal proteins, such as neural-cadherin (N-cadherin), fibronectin, and vimentin [23]. The loss of epithelial markers during EMT causes the nuclear translocation of β-catenin, activating the NF-kB pathway and thereby inducing the expression of different matrix metalloproteases (MMPs) [24]. Upon secretion of proteolytic enzymes, MMPs may degrade almost all the components of the extracellular matrix (ECM), such as collagens and laminin, influencing cell proliferation, migration, and adhesion [25]. In CRC, the EMT pathway has been associated with the increased expression of several MMPs, such as MMP2/7/9, by both in vitro studies in CRC cell lines and integrated multi-omics investigations [26,27]. Therefore, EMT may lead to the epithelial cell–cell junction disruption, loss of apical–basal polarity and cytoskeleton remodeling. Meanwhile, mesenchymal characteristics such as enhanced migratory capacity, invasiveness and elevated resistance to apoptosis are acquired [28]. Accordingly, recent studies showed that metastatic CRC patients with CTCs expressing EMT-related genes have worse progression free survival (PFS) and shorter overall survival (OS), suggesting the possibility of exploiting EMT-CTCs as prognostic factors in CRC [29].

CLMs spread can be enhanced by angiogenesis process. It has been shown that, in the presence of the primary tumor, the liver parenchyma adjacent to the liver metastases provides an angiogenic prosperous environment for metastatic tumor growth [30]. However, recent studies demonstrated that CLMs can also metastasize via a non-angiogenic process mediated by EMT, known as vessel co-option [31]. Interestingly, CRC cells can hijack pre-existing vessels of the host liver when conditions are not favorable to form new ones [32]. In CLMs, high levels of EMT markers have been detected in co-opted tumors compared to their angiogenic counterpart [33]. Indeed, Rada et al. recently demonstrated that EMT enhances CRC cells infiltration within liver parenchyma and the motility in hepatocytes, which displaced at the edge of tumor nests to allow the occupation of their space by cancer cells [34]. Another crucial step for metastatic invasion is represented by the disruption of blood vessels integrity [35]. The resulting altered vascular permeability allows tumor cells intravasation in the blood stream and subsequent extravasation in the metastatic site (Figure 1). The adherence of circulating CRC cells to the sinusoidal endothelial cells (SECs) of the liver is a crucial process involved in liver invasion and occurs upon selectins binding [36]. Rigorous blood vessel integrity is normally maintained by adherent junctions, whose basal organization is dictated by vascular endothelial cadherin (VE-cadherin), enforced by p120-catenin (p120) and spans the plasma membrane [37,38]. In such scenario, Dou et al. demonstrated that CRC cells that underwent EMT can interact with endothelial cells to increase vascular permeability. Specifically, EMT-CRC cells produced exosomal miR-27b-3p, which once transferred to endothelial cells, enhanced blood vessel permeability, thus facilitating cancer cell extravasation in the metastatic site by targeting VE-cadherin and p120 [39].

Once in the circulation, CTCs are subjected to a harsh selective pressure imposed by shear forces within the vessels but also the evasion of immune surveillance by the numerous immune cells in the plasma [40]. Accordingly, recent studies in mouse models showed CTCs survive in blood circulating for a short time, before being rapidly eradicated [41]. In order to survive in the bloodstream and form a niche at secondary sites, CTCs exploit EMT that activates survival pathways (such as activation of Akt, PI3K or EGFR pathways), that enable EMT-shifted cells to better resist apoptosis [19]. Moreover, EMT-mediated overexpression of the cell–cell junction marker plakoglobin may lead to generation of CTCs clusters, comprised of tumor cells with or without other non-malignant cell types, such as mesenchymal cells, epithelial cells, immune cells, platelets, and cancer-associated fibroblasts, that can contribute to the survival and metastatic advantages of CTC clusters [42]. Furthermore, the development of strong intercellular bonds allows invading cancer cells to resist the forces of plasma flow and circulating blood cells, when they adhere to hepatic sinusoids. Tumor cell adhesion and stabilization under the hydrodynamic conditions of blood flow are correlated to the expression of multiple signaling molecules, such as focal adhesion kinase, paxillin, and cytoskeletal proteins, such as actin or microtubules [43].

Subsequently, tumor cells start to interact with the extracellular matrix (ECM) of the invaded tissue, exploiting integrins, essential receptors able to stabilize the tumor cell adhesion [44]. Once disseminated to a distant site, cancer cells are believed to regain epithelial properties in a reverse process, referred to as mesenchymal-epithelial transition (MET), characterized by loss of migratory ability, with cells adopting an apical–basal polarization and expressing junctional complexes [45]. MET leads to the inhibition of cancer cells migration and induces their proliferation, leading to the growth of the new tumor. However, the definition of EMT has now been broadened, based on many new observations of intermediate hybrid epithelial and mesenchymal phenotypes, referred as partial EMT state [46]. These intermediate states that easily induce or reverse EMT process reflect the delicate balance of transcriptional drivers and suppressors of EMT and offer a more dynamic interpretation of the fluidity and plasticity of this phenomenon. Indeed, a recent study performed on CRC tissue samples showed the involvement of partial EMT at the invasive tumor front and the involvement of partial MET in both lymph node and liver metastases, based on the expression patterns of the miR-200 gene family in these critical locations [47]. These results highlighted that CRC cells undergo EMT at the invasive front of primary site along with their ability to recapitulate the phenotype of the primary tumor in the metastatic site.

## 3. The Molecular Mechanisms of EMT in CRC

One of the most distinguishing features for the establishment of an EMT phenotype is the overexpression of mesenchymal markers and deregulation of structural adhesion proteins. Among the aforementioned mesenchymal proteins, the most relevant ones induced in CRC during EMT are N-cadherin, vimentin, and fibronectin [21]. N-cadherin is a calcium-dependent transmembrane glycoprotein that mediates cell–cell adhesion, whose aberrant expression has been observed in many cancers because of it is closely related to cancer cells transformation and invasiveness [48]. Instead, vimentin is a major constituent of the intermediate filament family of proteins, whose physiological role is to maintain cellular integrity and provide resistance against stress in normal mesenchymal cells [49]. In CRC, the mesenchymal markers, N-cadherin and vimentin, have been shown to drive malignant progression of tumor cells and to correlate with metastasis development and a worse OS in patients [50,51]. Fibronectin is a soluble protein, part of the extracellular matrix (ECM), which in physiological conditions plays a role in wound healing [52]. Downregulation of fibronectin has been shown to inhibit colorectal carcinogenesis by suppressing proliferation, migration, and invasion [53]. Furthermore, fibronectin has been demonstrated to promote tumor cells growth and drugs resistance through a CDC42-YAP-dependent signaling pathway in CRC [54].

On the counterpart, E-cadherin, essential for the maintenance of adherent junctions, is fundamental for the physical integrity and polarization of epithelial cells [55]. Notably, loss of E-cadherin expression has been associated with poor prognosis in stage III CRC patients [56]. E-cadherin expression can be regulated at different levels in response to vary induction signals, including transcriptional repression [57], promoter methylation [58], as well as protein phosphorylation and degradation [59]. Indeed, the discovery that transcriptional repressors of E-cadherin contribute to invasion and metastasis has strengthened the evidence for the importance of the EMT in tumor progression.

EMT is modulated at different levels by epigenetic modifications, transcriptional control, alternative splicing, protein stability, and subcellular localization [21]. The transcriptional control of EMT is mainly driven by three groups of EMT-inducing transcription factors: Snail, Zeb, and Twist families (Table 1). Snail family is characterized by zinc finger transcription factors, all of which bind to a common binding motif known as the E-box and comprise SNAIL1 and SNAIL2 (also known as SLUG) genes [60]. Besides their role to repress E-cadherin expression, Snail transcription factors are involved in promoting the expression of mesenchymal genes, such as vimentin, N-cadherin, and fibronectin [61]. Indeed, Snail1 expression in CRC is associated with tumor progression and metastasis because it leads to the silencing of E-cadherin expression and to the induction of EMT [62]. Franci et al. showed that about 77% of colon cancer samples display Snail1 immunoreactivity both in activated fibroblasts and in carcinoma cells that underwent EMT, suggesting that the presence of Snail1 immunoreactive cells may be exploited as prognostic marker in patients with colon cancer [63,64]. Instead, the Zeb family comprises the 2-handed zinc finger/homeodomain proteins Zeb1 and Zeb2, which are especially overexpressed in CRC [65,66]. These transcription factors regulate EMT by binding E-box on E-cadherin promoter region and by upregulating LAMC2 mesenchymal gene to promote tumor invasion [67]. A recent study showed that the loss of a circadian gene, named Timeless, was able to induce EMT and E-cadherin downregulation in CRC cell lines, via a Zeb1-dependent mechanism [68]. Furthermore, it has been described that transcription factor Zeb1 causes severe alterations in the expression patterns of chromatin-modifying enzymes in CRC [69]. This study reported that the Zeb1-mediated upregulation of histone methyltransferase SETD1B stabilized Zeb1-mediated EMT through a positive feedback loop between Zeb1 and SETD1B, thus each protein can reinforce the activity/expression of the other. Lastly, Twist family, which comprehend Twist1 and Twist2 transcription factors, activate N-cadherin promoter and switch on mesenchymal markers such as N-cadherin and fibronectin, leading to E-cadherin-mediated cell–cell adhesion is lost and thereby promoting EMT [70]. Twist mediates an aggressive phenotype in human CRC cells. Indeed, a recent experiment performed on human CRC cell lines showed that Twist overexpression triggers EMT by E-cadherin downregulation and enhances tumor migration and invasion [71]. In addition, Twist-overexpressing CRC cells were more chemo-resistant to the drug oxaliplatin than control cells (Figure 2) [72].

Multiple miRNAs are thought to even govern EMT. In CRC, it has been found that miR-17-5p overexpression in tumor cell lines significantly decreased vimentin mRNA and protein expression, cell migration, and invasion, whereas downregulation of miR-17-5p in CRC cell lines increased vimentin protein expression, cell migration and invasion in vitro [73]. In addition, the authors’ findings suggested that overexpression of miR-17-5p inhibited liver metastasis in an intra-splenic injected mouse model. In another study, it has been found that miR-566 overexpression markedly increased the E-cadherin expression and inhibited the levels of vimentin and N-cadherin in several CRC cell lines (SW480, SW620, LoVo, HT29 and Caco-2) [74]. Therefore, miR-566 overexpression inhibited PSKH1 gene, suppressing cell proliferation, whereas inhibition of its expression enhanced cell survival and proliferation. The correlation between altered expression of specific miRNAs and CRC insurgency have triggered the examination of microRNAs as urinary noninvasive biomarkers, alternatively to invasive colonoscopy to detect early stages of the disease [75].

## 4. The Relationship between EMT and TGF-β Signaling Pathway in CRC

The transforming growth factor-β (TGF-β) is multifunctional cytokine that plays a role not only in the regulation of EMT, but also in the survival, development and differentiation of almost all cell types and tissues [76]. In cells of epithelial- and endothelial-origins, TGF-β also is a powerful suppressor of cell growth and proliferation [77]. Indeed, in the colon tissue, TGF-β signaling regulates the growth of normal cells in the colonic crypt and villi [78]. However, CRC can evade the tumor suppressing effects of TGF-β pathway, which represents one of the most commonly altered pathways in human cancers [79]. TGF-β signaling exists in three isoforms (TGFβ1, TGFβ2, TGFβ3) and it is mainly divided into two subfamilies: the TGF-β-activin-nodal subfamily and the bone morphogenetic protein (BMP) subfamily. TGF-β pathway is triggered via transmembrane serine/threonine kinase TGF-β type I receptors (TGF-βRI or ALK5) and TGF-β type II receptors (TGF-βRII). Upon TGF-β binding, TGF-βRII recruits and phosphorylates the cytoplasmatic domain of TGF-βRI, leading to the phosphorylation and activation of downstream transcription factors, SMAD2 and SMAD3 in the TGF-β-activin-nodal subfamily and SMAD1/5/8 in BMP subfamily [80]. In both cases, these activations allow them to bind SMAD4, generating SMAD complexes that translocate into the nucleus and bind DNA in a cell-specific manner, thus regulating the transcription of a multitude of TGF-β-responsive genes, including transcription factors belonging to the Snail family (e.g., Snail, Twist, or ZEB1), or of STAT3 [81,82]. The activation of these transcription factors elicits EMT-gene expression and ultimately promotes the prolonged induction of EMT via DNA methylation-mediated silencing of E-cadherin expression, as well as the upregulation of mesenchymal markers [83]. Among the numerous mesenchymal EMT-associated genes upregulated by TGF-β, there are N-cadherin, vimentin, and fibronectin, but also β3 integrin and several matrix metalloproteinases, such as MMP-3 and MMP-9 [84]. Indeed, TGF-β is often used in the cell culture to induce EMT and metabolic reprogramming in various epithelial cells [85,86]. Alternatively, activation of noncanonical TGF-β signaling, such as mitogen-activated protein kinase (MAPK), phosphoinositide 3-kinase (PI3K)/Akt and Rho/Rho-associated protein kinase (ROCK) pathways, also works with TGF-β in its regulation of EMT [83].

Altered functions of TGF-β lead to different gene expression patterns contributing to the development of oncogenic signaling and increasing the invasiveness ability of cancer cells [87]. Indeed, the initiation of oncogenic signaling boosted by TGF-β converts the regulation of physiological EMT in normal epithelial cells to pathological EMT in their malignant counterparts [84]. Loss of SMAD proteins represents one of the leading causes, and almost 25% of patients affected by CRC display a mutation in the SMAD4 protein complex [88]. Notably, the consensus molecular subtype (CMS) 4 of CRC displays the downregulation of SMAD4 [89]. Instead, SMAD4 upregulation suppresses invasion and restores the epithelial phenotype in the SW480 CRC cell line. Knockdown of SMAD4 led to increased levels of endogenous TGF-β cytokines, which upregulated TGF-β signaling and induced EMT [90]. SMAD4 is also a central component of the BMP signaling pathway, implicated in CRC pathogenesis. Hence, it has been shown that loss of SMAD4 alters BMP signaling and promotes CRC metastases via activation of Rho and ROCK pathways, leading BMP signaling to switch from tumor suppressive to metastasis-promoting function [91]. Contrarily, a recent study showed a SMAD4-independent EMT pathway in CRC, in which two epithelial SMAD4^mut^ CRC cell lines were able to acquire mesenchymal characteristics and regulate EMT marker genes in response to Snail1 induction, with phenotype independent from TGF-β and BMP receptor activity. These results suggested that there might be alternative transcription factors taking over the gene regulatory functions of SMAD4 during EMT in CRC [92].

TGF-β signaling has been shown to possess a dual role in the tumor microenvironment (TME), known as the “TGF-β paradox” [93]. In early-stage tumors, the TGF-β pathway acts as a tumor suppressor by inducing apoptosis, triggering cell cycle arrest, and thus inhibiting the proliferation of cancer cells [94]. In contrast, in late-stage tumors, it has pro-tumoral effects by modulating genomic instability, epithelial-mesenchymal transition (EMT), neo-angiogenesis, immune evasion, cell motility, and metastasis [95,96]. In this context, in CRC with high microsatellite instability (MSI-H subtype), TGF-βRII mutations interfere with TGF-β-induced EMT and therefore reduce the migratory and invasive capabilities of CRC cells, providing a better prognosis than microsatellite-stable CRCs [97,98]. On the other hand, the CMS4 subtype, with reduced expression of SMAD4, has been associated with poor OS [89,99]. In such a scenario, the bidirectional activity performed by TGF-β signaling in different CRC subtypes reflects the complexity of this signaling pathway in this tumor. Indeed, TGF-β signaling mutations enhance EMT and, subsequently, tumorigenesis and metastases in the CRC-CMS4 subtype, while TGF-βRII mutation impairs EMT and provides a better prognosis [100].

## 5. The Role of Tumor-Associated Macrophages (TAMs) in CRC Progression

The tumor microenvironment (TME) provides an essential dynamic niche with a key role in cancer initiation and progression [8,101,102]. In CRC, the microenvironment is composed of stromal cells and immune cells such as granulocytes, lymphocytes, and tumor-associated macrophages (TAMs), which affect tumor immune-suppression and inflammation [103,104]. Among the immune cells, TAMs represent the dominant cell type in TME and exhibit different functional polarization, known as M1 or M2 phenotype, in response to various stimuli from both tumor and stromal cells [105,106]. Different methodological approaches have allowed for highlighting the heterogeneity of TAMs in terms of function, polarization, and tissue localization [107]. Within the liver, these technologies include gene expression profiling [108], morphological identification [109], and evaluation of TAMs-specific markers [110]. Moreover, high dimensional analysis exploiting single-cell and spatial genomics technologies, such as single-cell RNA sequencing and mass cytometry by time-of-flight (CyToF), have allowed us to assess TAMs heterogeneity at an unprecedented resolution [111].

The classical M1 polarization is induced by recognition of pathogen-associated moieties, such as lipopolysaccharides (LPS) and Interferon gamma (INF-γ), with a key role in the innate response against pathogenic infection. Indeed, M1 macrophages are mainly involved in proinflammatory responses by producing proinflammatory cytokines (such as IL-12, IL-23) and chemokines (such as CXCL9 and CXCL10) [112,113]. M1 polarization can be identified by overexpression of CD80, CD86 and CD16/32 markers [114]. In contrast, M2 macrophages, which are induced by IL-4 and IL-13, exert a more anti-inflammatory response This population is characterized by elevated expression of arginase-1 (Arg-1), mannose receptor (CD206) and by secretion of anti-inflammatory factors (IL-10), chemokines (CCL17, CCL2) and matrix metalloproteinase 9 (MMP9) [114,115]. The majority of intra-tumoral macrophages has been shown to exhibit an M2 phenotype and to be correlated with poor prognosis in several tumors, including CRC [116,117]. In such scenario, flourishing literature demonstrated that in response to signals released from cancer cells, adaptive immune cells, B cells, fibroblasts, and macrophages themselves, such as IL-10, CCL2/3/4/5/7/8 and CXCL12 (colony stimulus factor 1 (CSF-1)), VEGF and interleukin-6 (IL-6), monocytes are recruited in the tumor niche and differentiate into the TAMs with an M2-like phenotype [118]. This population creates an environment that supports tumor growth and metastases by promoting tissue remodeling, angiogenesis, and secreting immunosuppressive cytokines, thus inhibiting innate and adaptive immune responses [112].

As mentioned above, TAMs play an important role in promoting tumor development and recurrence, as well as in the efficacy of anticancer strategies [8]. Despite the dichotomous M1/M2 classification, recent single-cell-resolution approaches have been useful in recognizing the heterogeneity of TAMs beyond the classic M1-like or M2-like phenotypes [102]. TAMs constitute a diverse macrophage population that shares features of both the M1 and M2 subsets. Therefore, TAMs plasticity could be associated with peculiar roles that macrophages exert in different cancers. With a particular focus on CRC, numerous studies have shown that a high density of macrophages is indicative of favorable outcomes in patients with stage III CRC [119]. The abundance of TAMs, particularly in CRC stage III metastatic lymph-nodes, might modify the efficacy of 5-fluorouracil chemotherapy, increasing CRC cell death and thus leading to better disease-free survival (DFS) [111]. Meanwhile, other data support the opposite findings. *Herrera* et al. reported that infiltration of CD163^+^ macrophages together with cancer-associated fibroblasts (CAFs) in CRC tissues was related to worse OS and PFS [120]. A recent study showed that a high density of macrophages correlates with worse DFS in patients who underwent chemotherapy for unresectable metastatic CRC after resection of the primary tumor. Indeed, TAMs were observed to induce chemoresistance by promoting malignant angiogenesis [121]. Furthermore, TAMs exert immunosuppressive roles in the CRC microenvironment. They recruit regulatory T cells (Tregs) by secreting the chemokine CCL2 and IL-10. TAMs suppress the antitumor immune response of T cells by metabolic starvation and inappropriately skew dendritic cells toward an immature and tolerogenic state [122]. In addition, immunosuppressive TAMs are characterized by high expression of immune-checkpoint molecules (such as PD-L1), causing T-cell exhaustion [123].

## 6. Mechanisms Exploited by TAMs to Regulate EMT in Colorectal Liver Metastasis (CLMs): A Focus on TGF-β Signaling Pathway

Despite the huge amounts of studies addressing the role and clinical relevance of TAMs in primary CRC, less is known concerning their role in CLMs, probably due to the different phenotypic profiles expressed by macrophages in the liver [124]. Kupffer cells (KCs) are tissue-resident macrophages localized within the lumen of the liver sinusoids. KCs perform phagocytic and cytokine secretion activities that allow them to eliminate circulating molecules and pathogens [125]. On the other hand, monocyte-derived macrophages are mainly resident in the portal triad, where they contribute to iron and cholesterol metabolism [126]. Therefore, TAMs with different morphologies and molecular fingerprints coexist in CLMs and correlate with clinicopathological variables [112]. Indeed, a recent study demonstrated that the morphology of tumor-associated macrophages (TAMs) correlates with prognosis in CLM patients. Specifically, while TAMs density did not correlate with survival in CLM patients, large (L)-TAMs, characterized by a bigger cell area and perimeter, were associated with statistically significantly worse prognosis and DFS, compared to small (S)-TAM ones [110].

The crosstalk between TAMs and EMT-CRC cells has been experimentally investigated by both in vitro and in vivo approaches. A co-culture assay was recently realized in vitro to evaluate the role of TAMs in CRC EMT, migration, and invasion [127]. This assay allows us to appreciate the changing morphology and gene expression profile of both cellular populations during their crosstalk, as well as to characterize the cytokines involved in cell–cell interactions by the analysis of the co-culture supernatants. Although in vitro metastasis models allow for the manipulation of each metastasis step, they do not provide a comprehensive analysis of the whole metastatic process [128]. In contrast, in vivo metastasis models may be more accurate in the representation of the metastatic process and can be genetically manipulated to mimic human cancer. In CLMs, in vivo experiments have been performed by realizing mouse xenografts, in which HCT116 CRC cell line and in vitro polarized TAMs were subcutaneously injected in 6–8-weeks-old nude mice [127]. Moreover, HCT116 xenografts can be coupled with macrophage depletion, such as by intravenous liposomal clodronate injection [129]. In this context, it is possible to study TAMs–CRC cells interactions in vivo, TAMs migration, and polarization, as well as how this crosstalk can change in the presence of specific microRNAs or interleukins that can be injected into the implanted tumor. However, the disadvantages of this model are the difficulty in assessing the contribution of the immune system in the metastatic process, because the various populations of immune cells may not properly function in the xenograft TME [128]. Moreover, xenografts do not show tumor heterogeneity or histopathologic and genetic characteristics of the original tumor. Therefore, additional research should be performed to develop more precise models to study TAMs and EMT-CRC cell interaction.

In CRC metastatic process, increasing evidence highlighted that TAMs are responsible for the induction of EMT because of the release of multiple factors, such as IL-6 and TGF-β [112]. Particularly, CD163^+^ TAMs have been observed to secrete IL-6, which, upon binding of IL-6 receptor (IL6R) on the cancer cell surface, phosphorylate STAT3 (pSTAT3) that translocated to the nucleus and regulates the expression of varying microRNAs, including miR-506-3p, which promotes EMT (Figure 3A). Indeed, E-cadherin gene expression was reduced in TAMs-mediated EMT, while the mesenchymal marker, vimentin, was upregulated in CRC cell lines. Therefore, the crosstalk between TAMs and tumor cells has been proposed to play an important role in inducing EMT in CRC and promoting EMT-mediated metastasis [127]. Moreover, IL-6 secretion by TAMs has been observed to induce chemoresistance through the activation of the IL6R/STAT3/miR-204-5p pathway in CRC cells [130]. M1 macrophages exhibited a potential to induce EMT in HCT116 and RKO CRC cell lines by secreting TNF-α and IL-1β [131,132]. Specifically, TNFα expression can improve Snail protein expression and nuclear localization through the AKT pathway, upregulating N-cadherin and fibronectin with a concomitant decrease in E-cadherin [132]. Il-1β secretion, instead, led to EMT of colon cancer cells with loss of E-cadherin, upregulation of Zeb1, and gain of the mesenchymal phenotype in CRC cell lines [131]. Recent evidence highlighted that CRC cells, through the NOTCH2/GATA3 pathway, undergo to EMT process and secrete IL-4, thus polarizing macrophages into an M2-like phenotype [133]. Another study showed that the expression of protein phosphatase of regenerating liver-3 (PRL-3) in CRC cells was able to activate the MAPK pathway in TAMs, thus leading to the release of IL-6 and IL-8 and inducing the EMT in cancer cells [134]. Furthermore, in vivo and in vitro studies showed that mesenchymal CRC cells obtained by Snail-induced EMT were able to secrete CXCL2, which promoted M2 macrophage infiltration and tumor cell metastasis [135]. These results suggest that not only macrophages play a role in EMT, but also tumor cells undergoing EMT could influence TAMs polarization. However, although most of the studies in CRC have focused on TAMs’ roles in promoting tumor metastasis and EMT, recent evidence showed that TAMs could inhibit EMT in patients affected by sporadic CRC, thus exerting a protective role against the development of metastases [136]. Strong CD68^+^ infiltration was reported to inhibit tumor burden, demonstrating that macrophages could have an important role in fighting against CRC cells with EMT traits.

TGF-β signaling pathway was highlighted to be involved in the crosstalk between TAMs and tumor cells in CLMs tumor microenvironment and to facilitate the induction of EMT in CRC cells [80]. It has been described that TGF-β secreted by TAMs activated the SMAD signaling pathway by binding to the TGF-β receptors, followed by the phosphorylated Smad2/Smad3 to form a complex with Smad4 and regulate transcription of Snail. Once TGF-β triggered EMT, Snail could repress the expression of E-cadherin, resulting in CRC metastasis [137]. Further evidence demonstrated that Collagen Triple Helix Repeat Containing 1 (CTHRC1) produced by CRC cells increased tumor burden and the number of CLMs nodules in mouse models by modulating macrophage polarization to M2 phenotypes through TGF-β signaling. Hence, CTHRC1 bound directly to TGF-βRII and TGF-βRIII in TAMs, stabilizing the TGF-β receptor complex and activating TGF-β signaling. Moreover, the inhibition of TGF-β signaling in macrophages through CTHRC1 monoclonal antibodies, coupled with PD-1/PD-L1 blockade, effectively led to the reduction of CLMs [138]. Interestingly, different studies showed that CTHRC1 overexpression has been associated with worse OS and DFS in CRC patients, driving the pathogenesis of the EMT process in CRC by activating the TGF-β pathway [138,139]. TAMs and the TGF-β pathways are also involved in the induction of CRC immune evasion [111]. In pediatric patients, it has been observed that CD163^+^ macrophages promoted the progression of colorectal polyps by inhibiting the local T-cell response through TGF-β production [140]. In another study performed on surgically resected CRC tissues, immunohistochemistry analysis showed that M2 macrophages induced immunosuppressive T-reg cell generation through activation of the TGF-β/SMAD signaling pathway (Figure 3B) [141]. Furthermore, increasing evidence suggests that TGF-β secreted by TAMs could play crucial roles in pathophysiological processes by regulating multiple microRNAs (miRNAs) [142,143]. In this context, TGF-β1 protein levels were found to be highly expressed in CRC tumor tissues and in vitro polarized macrophages. TAMs, obtained by in vitro co-culturing of macrophages with conditioned medium from CRC cells, downregulated the expression of miR-34a in tumor cells by secreting TGF-β1, inducing VEGF upregulation and thereby promoting cell proliferation and invasion of CRC cells [142]. TGF-β1 upregulation has been observed in vessel co-opting CLMs as well. Indeed, recent evidence showed that the TGF-β1 pathway may act as a mediator contributing to hepatocyte displacement during CLM development [34]. Although the role of TAMs remains unclear in this process, recent research showed that M1-macrophages are predicted to interact with pericytes via TGF-β signaling in vessel-co-opted tumors [144]. However, other studies found that the expression of TGF-β1 and SMAD4 in the cytoplasm, as well as the presence of TGF-βRII in the membranes of CRC cells, were associated with lower levels of CD68^+^ macrophages. Hence, the main components of the TGF-β1 signaling pathway (e.g., TGF-β1, SMAD4, TGF-βRII) could have an immunosuppressive effect via inhibition of macrophage recruitment [143]. Therefore, although TAMs can be one of the sources of TGF-β expression, the effects of TGF-β signaling on TAMs in CRC are not fully understood and requires further investigation.

Among the different mechanisms correlating TAMs and EMT, recent studies showed the potential use of anti-TGF-β strategies to impair CRC development, particularly in its late stages [145]. However, anti-TGF-β therapy alone is insufficient to mediate antitumor immunity in CRC. To overcome this problem, studies on the combination of other biological agents or irradiated tumor vaccine with anti-TGF-β treatment showed a reduction in TGF-β-induced EMT and correlated CRC metastasis [146]. Therefore, the combination therapy of chemotherapy/radiotherapy/targeted therapy with anti-TGF-β might be developed to achieve enhanced antitumor efficacy by regulating the tumor microenvironment. For these reasons, the exploration of the mechanisms of TGF-β signaling to develop TGF-β-based combination therapies might be very crucial for the development of new therapeutic applications in CRC and CLM [147].

## 7. Concluding Remarks and Future Perspectives

In this review, we discuss the involvement of EMT in the development of colorectal liver metastases (CLMs) and focus on the different signaling pathways exploited by TAMs to promote EMT in CRC, particularly describing the TGF-β signaling pathway.

Curative treatment of CLMs relies on surgical resection and systemic chemotherapy, providing the greatest chance for long-term survival. The 5-year survival after surgical resection is approaching 50%, with a 5-year disease-free survival of 25% [148,149]. Unfortunately, only 10–20% of patients are candidates for curative surgery. Instead, unresectable patients display 3-year survival of only 15% and are left with palliative chemotherapy as their only treatment option [150,151]. Fluoropyrimidine-based combinations (FOLFOX and FOLFIRI) with or without target therapy using anti-vascular endothelial growth factor (VEGF) or anti-epidermal growth factor receptor (EGFR) inhibitors represent the systemic treatments for CLM [152]. This combined approach offers the possibility to undergo hepatic resection even to those patients who would not have been considered for surgery until a few years ago. However, CLM patients display varying degrees of response to therapy, and research over the last decade has aimed to characterize the invasion of immune cells into the TME to stratify patient outcomes [153]. Anti-PD-1 or anti-PD-L1 blocking antibodies have shown significant results in CRC with microsatellite instability (MSI) [154]. Moreover, the neutralization of IL-10 effects in human CLMs has shown therapeutic potential as only treatment and to augment the function of administered CAR-T cells [155]. Nevertheless, the characterization of the immune landscape in CLM patients is complicated by the profound heterogeneity of tumor lesions across patients and the frequent neoadjuvant treatments of CLM patients, which could potentially affect the type of immune infiltrate [153].

Tumor-associated macrophages (TAMs) are essential players in CRC metastatic process, exerting a crucial role in EMT [156]. Upon secretion of various cytokines and other signaling molecules, such as exosomes, macrophages can crosstalk with CRC cells, promoting the EMT of tumor cells. In turn, mesenchymal tumor cells have been observed to enhance the recruitment of TAMs to the tumor site and promote their M2 polarization by secreting IL-10 and IL-4 [157]. In recent years, strategies targeting TAMs, including TAMs depletion, reprogramming, and inhibition of TAMs recruitment, have been investigated [111]. However, due to the significant heterogeneity of TAMs in the liver and in regulating tumor metastasis, results obtained from clinical studies not always were satisfying [158]. Therefore, it is necessary to further investigate the more unknown mechanisms by which TAMs promote CRC metastases, including EMT. Indeed, targeting a single EMT receptor is unlikely to be effective because of the redundant nature of several pathways and the biological features of EMT-transformed cells, such as increased cell mobility, invasiveness, and chemoresistance, further complicating drug development [159]. A wide range of targets associated with EMT is required to be elucidated in the future to overcome therapy resistance. Novel therapies have been proposed in oncology targeting the TGF-β signaling pathway, exploited by TAMs to promote EMT and responsible for resistance to conventional therapies in CRC [160]. Several TGF-β pathway antagonists have advanced to clinical trials and demonstrated acceptable safety profiles and significant therapeutic efficacy in cancer patients. In addition, TGF-β agonists may be exploited in patients resistant to conventional therapies [161]. Chemotherapy drugs ginsenoside Rb2 and tanshinone II A displayed therapeutic effects acting as inhibitors of TGF-β-induced EMT and angiogenesis, respectively [146,162]. Among the numerous pharmacological approaches targeting TGF-β that have undergone preclinical and clinical stages, there are neutralizing antibodies, TGF-β inhibitors, ligand traps, antisense oligonucleotides, and vaccines [145]. In CRC, preclinical trials have shown that the combination with TGF-β inhibitor Galunisertib, a selective inhibitor of TGFβRI, was able to enhance the efficacy of chemotherapy and radiotherapy [163]. Coadministration of TGF-β inhibitors and anti-PD-L1 antibodies displayed effective response in CRC patients by promoting CD8^+^ T cells penetration into the tumor [164]. Moreover, it has been shown that dual blockade of TAM recruitment and TGFβ signaling significantly augments the therapeutic efficacy of chemotherapy via suppressing PD-L1 expression in metastatic CRC [165]. However, despite the suitable results obtained in the ongoing clinical trials, the mechanisms underlying TGF-β mediated EMT promoted by TAMs remains unclear in CLMs patients. For this reason, further investigations in this field may provide new therapeutic strategies for fighting CLMs, which are still responsible for most of the deaths in CRC patients.

## Figures and Tables

**Figure 1 ijms-23-16197-f001:**
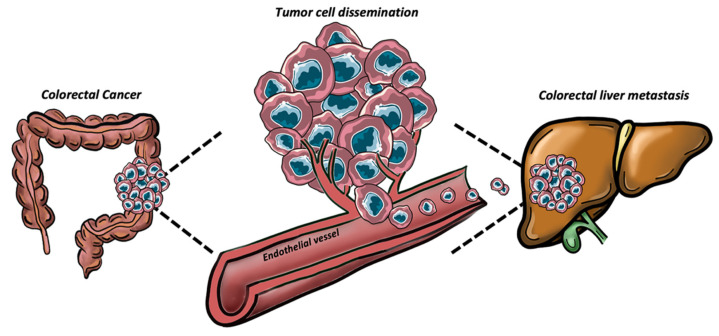
Representative image of the tumor cell dissemination process from the primary site (colon) to the metastatic one (liver).

**Figure 2 ijms-23-16197-f002:**
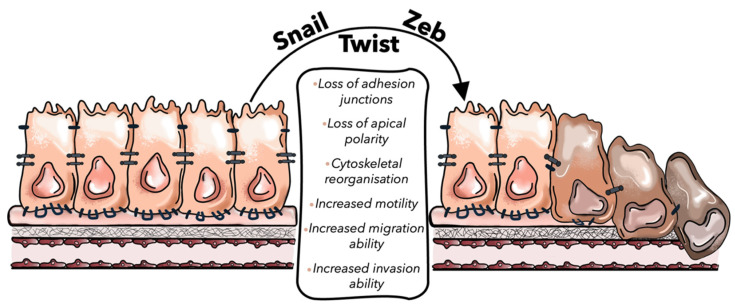
Representative image of the EMT process. Snail, Twist, and Zeb, EMT-promoting transcription factors (EMT-TFs), promote the loss of cellular epithelial characteristics, such as the disassemble of cell–cell junction, the defect of apical–basal polarity and cytoskeletal reorganization, supporting the acquisition of a mesenchymal phenotype of tumor cells.

**Figure 3 ijms-23-16197-f003:**
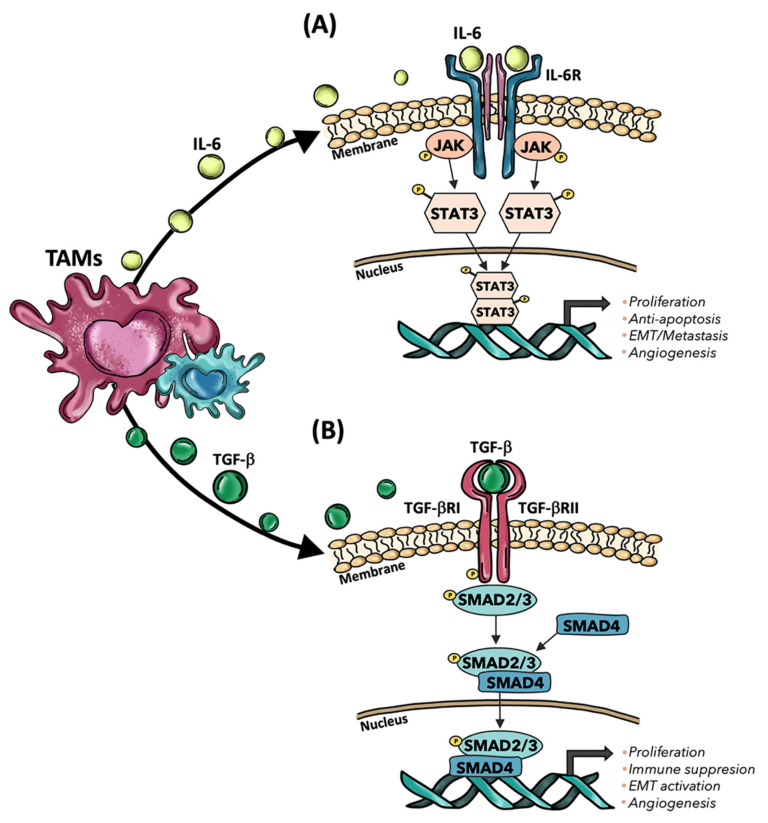
Mechanisms exploited by TAMs to induce cancer progression in CLMs. (**A**) TAMs release IL-6, which then interacts with IL-6R, subsequently activating the JAK-STAT pathway that regulates the expression of genes involved in tumor progression. (**B**) TGF-β secreted by TAMs interacts with TGF-βI and TGF-βII receptors, leading to phosphorylation of SMAD2/3 and subsequent generation of a complex with SMAD4, which finally leads to the expression of cancer-related genes. TAMs: Tumor-associated macrophages; IL-6: Interleukin-6; IL-6R: Interleukin-6 receptor; TGF-b: Transforming growth factor beta; TGF-βR: TGF-β receptor; JAK: Janus kinase; STAT3: signal transducer and activator of transcription 3; SMAD: Suppressor of Mothers against Decapentaplegic.

**Table 1 ijms-23-16197-t001:** EMT transcription factors and their role in EMT mechanisms to promote CLM development.

EMT-TFs	Members of TF Family	Mechanisms in CLM-Related EMT	References
** *Snail* **	*Snail1, Snail2* *(SLUG genes)*	▪ *E-cadherin downregulation* ▪ *Vimentin, N-cadherin, and fibronectin mesenchymal genes upregulation*	*[60,61]*
** *Zeb* **	*Zeb1, Zeb2*	▪ *E-cadherin downregulation* ▪ *LAMC2 mesenchymal gene upregulation* ▪ *Severe alterations in the expression pattern of chromatin-modifying enzymes*	*[67,68,69]*
** *Twist* **	*Twist1, Twist2*	▪ *N-cadherin and fibronectin upregulation* ▪ *Loss of E-cadherin-mediated cell–cell adhesion* ▪ *Tumor migration and invasion* ▪ *Possible induction of chemoresistance to the drug oxaliplatin*	*[70,71,72]*

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
