# Peer review of "Relationship between Epithelial-to-Mesenchymal Transition and Tumor-Associated Macrophages in Colorectal Liver Metastases"

_ijms, 2022, doi:10.3390/ijms232416197_

Round 1

Reviewer 1 Report

The review entitled „ Relationship between epithelial-to-mesenchymal transition and tumor-associated macrophages in colorectal liver metastases” by Gazzillo et al. summarizes an interesting and important issue. The manuscript is well written and comprises all relevant issues. Furthermore, the authors have included a number of descriptive figures to illustrate the topic.

Some points might be addressed:

1. The authors describe the role of tumor-associated macrophages in CRC progression very nice. I suggest including a paragraph on how TAMs can be identified in the tissue. Moreover, is there any possibility to differ macrophages with M1 andM2 polarization?

2. The role of EMT in tumor progression is illustrated in Figure2. I suggest including a table in which the effects of each factor of EMT on the development of CRLM is summarized e.g., the different factor is Zeb of Twist family. Similar, this might be included for TAMs and their subpopulations.

3. The authors state that the prognosis of patients with CRLM is still poor. I suggest distinguishing a little bit between those patients with resectable and those with irresectable disease. In fact, the prognosis differs a lot and is much better in those who can undergo surgical resection compared to patients treatable with systemic therapy only.

Author Response

REPLY TO THE REVIEWERS’ COMMENTS

The authors of the manuscript entitled: “Relationship between epithelial-to-mesenchymal transition and tumor-associated macrophages in colorectal liver metastases” have revised the review according to the comments of the Reviewers. They would like to thank the Editors and the involved Reviewers for the effort and time dedicated to giving feedback for their manuscript. They sincerely appreciate the suggestions and comments, which helped to improve the quality of their paper. Please find the changes in the revised version of the manuscript marked up using the Word “Track Changes” function.

Reviewer #1: The review entitled „Relationship between epithelial-to-mesenchymal transition and tumor-associated macrophages in colorectal liver metastases” by Gazzillo et al. summarizes an interesting and important issue. The manuscript is well written and comprises all relevant issues. Furthermore, the authors have included a number of descriptive figures to illustrate the topic. 

Reply: We thank the reviewer for the positive comments.

The authors describe the role of tumor-associated macrophages in CRC progression very nice. I suggest including a paragraph on how TAMs can be identified in the tissue. Moreover, is there any possibility to differ macrophages with M1 andM2 polarization? 

Reply: We thank the reviewer for this suggestion. The text of paragraph 5 has been revised to provide an overview of the methodologies exploited to identify the heterogeneity of TAMs in liver tissue. Moreover, a brief description of the different polarization markers was added to the same paragraph to make the reader aware of the possibility to differ M1 and M2 macrophages according to their unique transcription profile. The changes have been implemented using the Word “Track Changes” function.

The role of EMT in tumor progression is illustrated in Figure2. I suggest including a table in which the effects of each factor of EMT on the development of CRLM is summarized e.g., the different factor is Zeb of Twist family. Similar, this might be included for TAMs and their subpopulations. 

Reply: We thank the reviewer for this suggestion. Table 1 was added to provide a schematic overview of the relationship between each EMT-related factor and the EMT mechanism involved in  the development of CLMs. Regarding TAMs subpopulations table, even if considered an important topic by the authors, due to space constraints and the comprehensiveness of the TAMs heterogeneity issue, it is not possible to realize it in detail. Indeed, the aim of this review is to give an overview of the latest literature findings on the selected topic (the role of TAMs in CLMs through TGF-beta induced EMT) giving the reader hints and the starting point to execute a deeper search. However, paragraph 5 briefly describes the main characteristics of TAMs heterogeneity and relevant papers that accurately dissect the properties of polarized macrophage population were included in the review (see references n°8 and n°112). The change has been implemented using the Word “Track Changes” function.

The authors state that the prognosis of patients with CRLM is still poor. I suggest distinguishing a little bit between those patients with resectable and those with irresectable disease. In fact, the prognosis differs a lot and is much better in those who can undergo surgical resection compared to patients treatable with systemic therapy only.

Reply: We thank the reviewer for this comment. The text of paragraph 7 has been revised to describe the suggested distinction between the prognosis of resectable and unresectable patients. The change has been implemented using the Word “Track Changes” function.

Reviewer 2 Report

This is an interesting, general, review about the role that tumor associated macrophages (TAMs) exert in the induction of EMT during colorectal liver metastasis development. The text is well balanced and offers an interesting overview. However, it is a bit too superficial, and it fails to go in-depth in some of the areas where it mainly focuses.

The idea of merging TAM and EMT-mediated liver metastasis is novel and rather original and given that not much is known about this interaction, I suggest that the authors dedicate some effort to delineate how specifically the TAM and EMT-CRC cells might interact at a molecular level. This could refer to activation of vascular co-option programs, de-differentiation, and various other aspects where an interaction could be foreseen. They could also discuss in greater detail the possibilities, and challenges, resulting from exploring these approaches at the experimental level. I believe that this would make the review much more attractive to readers by providing an original viewpoint for investigation.

Major comments

1.- “Simple summary” section should be expressed is plain words. The “Abstract” section is even clearer to the general audience and in my opinion, should be the other way round.

2.- Authors should discuss in detail several processes at the molecular level such as the process of vessel co-option as a fundamental step of EMT-mediated CRC metastasis. A dedicated section entertaining these concepts should be added

3.- Authors should discuss in greater detail the possibilities, and challenges, resulting from exploring this interaction (TAM and EMT-CRC cells) at the experimental level.

4.- A short section devoted to therapy should be added to make the review more interesting to biomedical audience.

5.- The figures should be clearer to the readers. Names are missing and the color patterns are a bit confusing. At least “Graphical abstract” and “Fig1” should be corrected accordingly avoiding misleading artistic effects and adding names to the main actors of the cartoon model.

Minor comments

1.- There are entire sentences copied from published papers. To avoid plagiarism issues, I suggest the use of applications to detect plagiarism (i.e. the free tool https://www.plagiarismchecker.co/) and rewrite the sentence accordingly.

Author Response

REPLY TO THE REVIEWERS’ COMMENTS

The authors of the manuscript entitled: “Relationship between epithelial-to-mesenchymal transition and tumor-associated macrophages in colorectal liver metastases” have revised the review according to the comments of the Reviewers. They would like to thank the Editors and the involved Reviewers for the effort and time dedicated to giving feedback for their manuscript. They sincerely appreciate the suggestions and comments, which helped to improve the quality of their paper. Please find the changes in the revised version of the manuscript marked up using the Word “Track Changes” function.

Reviewer #2: This is an interesting, general, review about the role that tumor associated macrophages (TAMs) exert in the induction of EMT during colorectal liver metastasis development. The text is well balanced and offers an interesting overview. However, it is a bit too superficial, and it fails to go in-depth in some of the areas where it mainly focuses. 

The idea of merging TAM and EMT-mediated liver metastasis is novel and rather original and given that not much is known about this interaction, I suggest that the authors dedicate some effort to delineate how specifically the TAM and EMT-CRC cells might interact at a molecular level. This could refer to activation of vascular co-option programs, de-differentiation, and various other aspects where an interaction could be foreseen. They could also discuss in greater detail the possibilities, and challenges, resulting from exploring these approaches at the experimental level. I believe that this would make the review much more attractive to readers by providing an original viewpoint for investigation.

 We thank the reviewer for the comments. The text of paragraphs 6 has been revised to describe the suggested interaction between TAMs and EMT at a molecular level and how this crosstalk can be explored experimentally in both in vitro and in vivo assays, also underlying pros and cons of these approaches. Moreover, the text of paragraph 2 has been revised to describe further processes involving EMT in metastatic spread. The change has been implemented using the Word “Track Changes” function.

1.- “Simple summary” section should be expressed is plain words. The “Abstract” section is even clearer to the general audience and in my opinion, should be the other way round.

 We thank the reviewer for this suggestion. Simple summary was revised to become clearer to the general audience. Meanwhile, Abstract section was accordingly modified to provide an in-depth description of the topic addressed by the review. The changes have been implemented using the Word “Track Changes” function.

2.- Authors should discuss in detail several processes at the molecular level such as the process of vessel co-option as a fundamental step of EMT-mediated CRC metastasis. A dedicated section entertaining these concepts should be added

We thank the reviewer for the suggestion. In paragraph 2, we added accordingly a dedicated section of the involvement of EMT in different molecular processes of CLM development. Specifically, we dedicated an entire section to vessels co-option, CTCs survival in blood circulation, and EMT-mediated expression of metalloproteinases involved in ECM remodeling during metastatic invasion. The changes have been implemented using the Word “Track Changes” function.

3.- Authors should discuss in greater detail the possibilities, and challenges, resulting from exploring this interaction (TAM and EMT-CRC cells) at the experimental level.

We thank the reviewer for this consideration. A dedicated section to the experimental approaches applicable in the exploration of TAMs and EMT-CRC cells crosstalk has been accordingly added to paragraph 6. The change has been implemented using the Word “Track Changes” function.

4.- A short section devoted to therapy should be added to make the review more interesting to biomedical audience.

 We thank the reviewer for the suggestion. A short section devoted to therapy has been reported in paragraph 7. Moreover, paragraph 7 provides an overview of the novel therapies that have been proposed for CLMs exploiting TAMs and their ability to trigger EMT through TGF-beta pathway. The change has been implemented using the Word “Track Changes” function.

5.- The figures should be clearer to the readers. Names are missing and the color patterns are a bit confusing. At least “Graphical abstract” and “Fig1” should be corrected accordingly avoiding misleading artistic effects and adding names to the main actors of the cartoon model.

We thank the reviewer for this suggestion. All the figures were revised to add missing names and to correct color patterns to be clearer to the reader. The changes have been implemented using the Word “Track Changes” function.

Minor comments

1.- There are entire sentences copied from published papers. To avoid plagiarism issues, I suggest the use of applications to detect plagiarism (i.e. the free tool https://www.plagiarismchecker.co/) and rewrite the sentence accordingly.

We thank the reviewer for the useful suggestion. We checked the entire review using the suggested application and we rewritten the sentences accordingly. The changes have been implemented using the Word “Track Changes” function.

Reviewer 3 Report

General comment     

I read with great attention the manuscript entitled " Relationship between epithelial-to-mesenchymal transition and tumor-associated macrophages in colorectal liver metastases" by Gazzillo et al.

The manuscript presents a review of the current knowledge about TAM-mediated epithelial-to-mesenchymal transition, focusing on the involvement of TGF- β signaling in this process.

The scope of the review is of interest and relevant. This manuscript is well written.

The topic is of great importance and as the authors state the mechanisms underlying TGF-β mediated EMT promoted by TAMs remains unclear in patients with colorectal liver metastases.

Although there are other reviews on this topic, the presentation of the most recent advances in this field is pertinent. In this regard, the authors could extend a little the recent results and current studies that were briefly mentioned in section 7. Concluding remarks and future perspectives.

Grammar and spelling errors should be reviewed wholly.

Author Response

REPLY TO THE REVIEWERS’ COMMENTS

The authors of the manuscript entitled: “Relationship between epithelial-to-mesenchymal transition and tumor-associated macrophages in colorectal liver metastases” have revised the review according to the comments of the Reviewers. They would like to thank the Editors and the involved Reviewers for the effort and time dedicated to giving feedback for their manuscript. They sincerely appreciate the suggestions and comments, which helped to improve the quality of their paper. Please find the changes in the revised version of the manuscript marked up using the Word “Track Changes” function.

Reviewer #3: I read with great attention the manuscript entitled " Relationship between epithelial-to-mesenchymal transition and tumor-associated macrophages in colorectal liver metastases" by Gazzillo et al.

The manuscript presents a review of the current knowledge about TAM-mediated epithelial-to-mesenchymal transition, focusing on the involvement of TGF- β signalling in this process.

 The scope of the review is of interest and relevant. This manuscript is well written. 

The topic is of great importance and as the authors state the mechanisms underlying TGF-β mediated EMT promoted by TAMs remains unclear in patients with colorectal liver metastases. 

Although there are other reviews on this topic, the presentation of the most recent advances in this field is pertinent. In this regard, the authors could extend a little the recent results and current studies that were briefly mentioned in section 7. Concluding remarks and future perspectives.

Grammar and spelling errors should be reviewed wholly.

We thank the reviewer for the positive comments. Paragraph 7 was extended to provide a wider overview of the current treatments for CLM and the future perspectives associated to TGF-beta targeted therapies for this disease. However, literature investigating this topic is still poor. Moreover, grammar and spelling errors were entirely revised. The change has been implemented using the Word “Track Changes” function.